# Causal Frameworks and Feature Discrepancy Loss: Addressing Data Scarcity and Enhancing Medical Image Segmentation

## Abstract

Data scarcity poses a significant challenge for deep learning models in medical imaging, particularly for training and generalization. Previous studies have demonstrated the efficacy of data pooling from various sources, facilitating the analysis of weak but significant correlations between imaging data and disease incidence. This approach is often constrained by strict data-sharing protocols among institutions, resulting in models reliant on external data sources. In this work, we address the issue of data scarcity by leveraging the available data for segmentation tasks across various medical imaging modalities. Based on our observation that samples with minimal foreground-background feature differences often demonstrate inadequate segmentation performance, we propose a causal-inspired foreground-background feature discrepancy penalty function, which improves feature separation and alleviates segmentation difficulties caused by homogeneous pixel distributions. The proposed feature discrepancy loss is mathematically grounded, with a lower bound defined by the negative logarithm of the Dice coefficient, suggesting that increased feature separation correlates with improved Dice scores. To further validate our approach, we introduce a novel ultrasound dataset for triple-negative breast cancer (TNBC), and we evaluate the method across three state-of-the-art segmentation architectures to demonstrate competitive performance. In addition, the results highlight the robustness of our method in mitigating performance decrease due to distribution shifts when new, differently distributed data batches are introduced.

## 1 Introduction

Medical imaging datasets frequently suffer from limited sample sizes, often due to budget constraints and strict study criteria, including specific genetic risks. This scarcity of images and diagnostic labels complicates the training of deep learning models. A significant issue arises from the risk of learning spurious correlations within the dataset, which results from the weak statistical signal of the disease derived from a limited number of samples Thompson et al. (2014). Moreover, disparities in data distributions hinder model generalization to real-world clinical settings. Despite progress in predictive analytics, the lack of quality data and data mismatch remain significant barriers Moyer et al. (2018). Semi-supervised learning and data augmentation help address the issue, though with varying effectiveness Chapelle et al. (2006). Pooling data from multiple sites, along with methods like covariate matching and meta-analysis, enhances model robustness and generalizability. Lokhande et al. (2022)

**Limitations of Data Augmentation in Medical Imaging.** Data augmentation techniques, such as rotations, flips, and crops, are often applied to imaging data to enhance model robustness by generating additional plausible data points Carmon et al. (2019). However, in medical imaging, these techniques often fall short of their objectives. For example, cropping or flipping brain images can disrupt the brain's inherent asymmetry,

yielding irrelevant results Akash et al. (2021). Deformations must be carefully applied to maintain clinical relevance. Recent studies suggest data augmentation offers limited benefit in tasks like semantic segmentation as it often fails to generate realistic variations in object boundaries and spatial relationships (Oliver et al., 2018; Goceri, 2023). Data pooling from multiple sites helps address data scarcity, but distributional differences complicate harmonization, limiting the effectiveness of augmentation techniques.

**Integrating Causal Reasoning in Medical Imaging.** Causal reasoning Pearl (2009) is crucial in tackling challenges like data scarcity and dataset disparity in medical imaging, especially in machine learning Bareinboim & Pearl (2016). By establishing causal links between medical images and annotations, researchers can improve data collection, annotation, and learning strategies, while also addressing biases Schölkopf et al. (2012). In cases where anti-causal relationships exist, traditional semi-supervised methods may fall short. Causal insights enable more efficient use of limited labeled data and help mitigate selection biases. Utilizing causal diagrams to formalize assumptions about data generation enhances model robustness and generalization to real-world clinical data, improving diagnostic tools and the effectiveness of augmentation techniques Castro et al. (2020).

**Challenges in Breast Cancer Imaging Due to Data Scarcity.** Breast cancer is the most common cancer among women and a leading cause of cancer-related deaths. In Algeria, there are over 14,000 new cases reported each year Lagree et al. (2021); aps (2020). This paper focuses on breast cancer as the primary disease type among the medical imaging datasets analyzed. Early detection is crucial for better treatment outcomes, aided by advancements in medical imaging technologies like mammography, ultrasound, MRI, and histopathology. Upon identifying suspicious lesions such as nodules Evain et al. (2021) or microcalcifications Touami & Benamrane (2021), biopsies are performed to confirm diagnosis and cancer stage. Recent strides in machine learning and deep learning have surpassed traditional methods like watershed and super-pixels, with deep learning models such as FCN Long et al. (2015), U-Net Ronneberger et al. (2015), and DeepLab Chen et al. (2014) demonstrating high efficacy in medical image segmentation. Models like AlexSegNet Singha & Bhowmik (2023), CellTranspose Keaton et al. (2023), and MMPSO

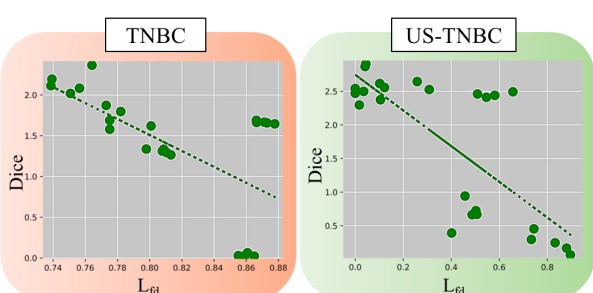

Figure 1: Correlation between Dice and $\mathcal{L}_{\mathbf{fd}}$ (foreground-background feature distance loss) is significant. This relationship is observed in the decoder layers of NucleiSegNet and the encoder layers of CMUNet. The correlation is evident in both ultrasound and histopathology images. In these cases, the foreground and background pixel distributions are homogenous. Consequently, distinguishing the foreground from the background is challenging due to their similarity. The paper introduces a new ultrasound dataset called US-TNBC.

Kanadath et al. (2023) further improve segmentation performance. MCFNet Feng et al. (2021) addresses spatial information but struggles with complex staining patterns. Multimodal approaches Dwivedi et al. (2022); Roy et al. (2024b); Chen et al. (2021), such as TGANet Tomar et al. (2022a), DTAN Zhao et al. (2024), and GRUNet Roy et al. (2024a), combine textual and spatial data to enhance segmentation. However, attention mechanisms and multimodal models fail to address homogeneous pixel distributions in ultrasound and histopathology images, leading to segmentation challenges. While breast cancer ultrasound datasets are available, a dedicated dataset for triple-negative breast cancer (TNBC), the most aggressive form, is lacking.

**Contributions.** This paper focuses on the segmentation task in medical imaging, a field that poses significant challenges in accurately delineating complex anatomical structures and pathologies. Effective segmentation is crucial for improving diagnosis, treatment planning, and patient outcomes in healthcare Malhotra et al. (2022). Our contributions are based on the observation that the Dice Score, a widely used metric for vali-

dating image segmentation quality, is correlated with the foreground-background feature distance produced by neural networks generating segmentation masks (see Figure 1). To leverage this insight, we propose the following: **(a)** a feature distance loss to enhance feature distinction, thereby reducing over- and under-segmentation in cases of homogeneous pixel distributions; **(b)** a demonstration that the negative logarithm of the Dice coefficient acts as a lower bound for the feature distance loss, ensuring improved Dice scores when optimizing for the feature distance score; **(c)** the introduction of a new ultrasound breast cancer dataset specifically for triple-negative breast cancer (TNBC); and **(d)** an approach to address dataset distribution shift issues when integrating datasets from multiple sources. We achieve state-of-the-art segmentation accuracies across five datasets and three architectures.

## 2 METHOD

### 2.1 CAUSAL STRUCTURE AND MODULARITY

A key challenge in medical image analysis is the scarcity of labeled data, largely due to the high cost of obtaining expert annotations or expensive laboratory tests. Understanding the variables that influence the data-generation process is essential for systematically addressing data scarcity. Causal reasoning provides a powerful framework for analyzing how these variables interact in the data-generation process. This approach examines cause-effect relationships between variables, which are represented as links or edges, forming a directed acyclic graph (DAG), also known as a causal diagram or structure. For further details, we refer the reader to Neuberg (2003).

In our study of medical images, $X$, and their corresponding segmentation ground truth targets, $Y$, it is essential to determine the causal relationship between them. The relationship between $X$ and $Y$ may be causal, represented as $X \rightarrow Y$, indicating a predicted effect from the cause. This suggests that $Y$ is mechanistically dependent on $X$, along with other factors and independent noise. Alternatively, the relationship may be anticausal, $Y \rightarrow X$, predicting the cause from the effect. Consistent with statistical machine learning principles, the task is to estimate $P(Y \mid X)$, irrespective of the direction.

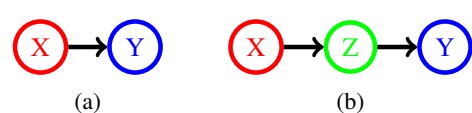

(a)        (b)

Figure 2: Causal diagram for the medical image segmentation problem. (left) 2a, the standard causal prediction model used for segmentation tasks. (right) 2b, a new mediator variable $Z$, aimed at addressing data scarcity challenges.

Segmentation tasks in histopathology datasets, such as TNBC Naylor et al. (2018), or ultrasound datasets like UDIAT Yap et al. (2017), necessitate manual segmentation of images $X$, with precise contouring of tumor or cell regions $Y$. This annotation relies on visual inspection and is affected by image content, resolution, and contrast. The annotator's understanding of tumor grade may influence the delineation of specific boundaries. Manually editing the segmentation masks does not change the original images. These factors indicate that segmentation adheres to a causal prediction model, that is, $X \rightarrow Y$.

**Axiom 1. (Modularity for X $\rightarrow$ Y):** *In the causal graph where X causes Y, intervening on X changes only the mechanism determining X, while the mechanism determining Y given X remains invariant.*

Axiom 1 indicates that $P(X)$ offers minimal information compared to $P(Y \mid X)$, implying that data augmentation and semi-supervised learning techniques are theoretically inadequate for resolving the data scarcity issue. A model trained on image-derived annotations will mainly reproduce the manual annotation process instead of predicting a pre-imaging ground truth, like the 'true' anatomy. While efforts to enhance data augmentation techniques for segmentation tasks continue Yellapragada et al. (2024), our approach emphasizes utilizing existing data to improve segmentation outcomes, as illustrated by the observations in Fig. 8 and Fig. 7.

## 2.2 Handling Data Scarcity through Causal Mediation

In the absence of data augmentation, we must utilize the existing samples in the dataset effectively. One strategy involves identifying underperforming samples and improving their performance. This method adheres to the Rawlsian principle of prioritizing the worst-off group of samples. Techniques like up weighting have demonstrated potential; however, they would be ineffective in this context, as identifying suitable weights necessitates access to a probability distribution that cannot be reliably estimated in data-scarce medical imaging situations. This paper addresses the issue through causal mediation, introducing intervening variables, $Z$, to mediate the relationship (see Figure 2). The mediator $Z$, obtained from the image $X$, functions as a differentiable proxy for $Y$.

**Proposition 2. (Mediation in Causal Prediction Model)**: *Given a causal diagram of the form $X \to Y$, introducing a mediator $Z$ to create the structure $X \to Z \to Y$, and assuming a strong correlation between $Y$ and $Z$, this results in*

- *(Conditional Independence): $(X \perp Y) \mid Z$*

- *(Preserved Modularity): $P(X) \perp P(Y \mid X)$*

- *(Functional Relationship): $P(Y \mid X) = \int P(Y \mid Z)P(Z \mid X)$.*

The relationship shown in equation 2 indicates that $P(Y \mid X)$ depends on $P(Z \mid X)$, as $Z$ mediates the complete effect of $X$ on $Y$. This indicates that an accurate determination of $P(Z \mid X)$ allows for precise estimation of $P(Y \mid X)$.

**Example 3.** *Consider $X \sim \mathcal{N}(0,1)$, where $\mathcal{N}$ denotes the normal distribution. Define $Z = aX + \epsilon_1$ and $Y = bZ + \epsilon_2$, where $\epsilon_1 \sim \mathcal{N}(0,1)$ and $\epsilon_2 \sim \mathcal{N}(0,1)$, and $a$ and $b$ are constants. Under these definitions, we have the following conditional distributions: $Z \mid X \sim \mathcal{N}(aX,1)$, $Y \mid Z \sim \mathcal{N}(bZ,1)$, and consequently $Y \mid X \sim \mathcal{N}(abX, 1 + b^2)$.*

The example demonstrates that $P(Y \mid X)$ is a function of $P(Z \mid X)$, as the mean of $Y \mid X$ (represented as $abX$) depends on the mean of $Z \mid X$ (which is $aX$). Moreover, conditional independence is preserved, as knowing $X$ provides no further information about $Y$ given $Z$.

## 2.3 Mediator as a Feature Distance Measure

The mediator variable $Z$ must capture causally relevant information for segmentation while discarding irrelevant or spurious correlations, encouraging generalization across diverse datasets while maintaining discriminative power for

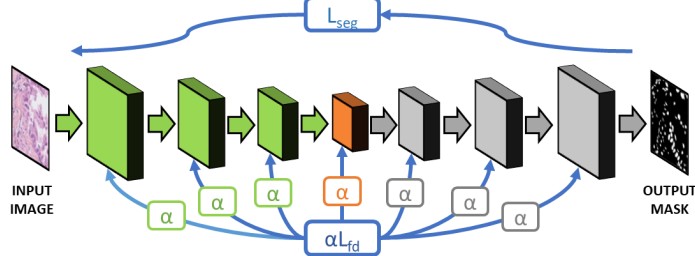

Figure 3: The proposed method shown with respect to UNet architecture. The green blocks are the Encoder layers, the grey blocks are the Decoder layers, and the orange block is the Bottleneck layer. Every layer is treated with the feature discrepancy loss ($\mathcal{L}_{\mathbf{fd}}$) with a learnable $\alpha$. $\alpha$ is trainable for all the layers and is unique for each layer.

foreground-background segmentation. In the UNET architecture Ronneberger et al. (2015), as in many other models, the feature map $F$ is represented by three dimensions: height, width, and channel. Access to ground truth masks or clustering methods during training helps identify indicators $\tilde{y}$ that differentiate between foreground and background features Sims et al. (2023). We demonstrate that increasing the distance between foreground and background features improves the estimation of $Z$. The corresponding distance penalty loss is formally defined as follows:

**Definition 4. (Feature Distance Loss):** *Let $F$ denote the features extracted from any network architecture and $\tilde{y}$ represent the indicator variables identifying foreground features. We define the channel-averaged foreground features as $F_g = \sum_k \left( \sum_{i,j} F[i,j,k] \otimes \tilde{y}[i,j,k] \right)$ and the channel-averaged background features as $B_g = \sum_{i,j} F[i,j,k] \otimes (1 - \tilde{y}[i,j,k])$, where $\otimes$ denotes element-wise multiplication. The feature distance loss is then given by*

$$\mathcal{L}_{\textit{fd}} = -\log \left( \|F_g - B_g\|^2 \right) \tag{1}$$

In the previous discussion, $F_g - B_g$ reflects the difference in foreground and background features. This penalization of feature differences helps the model identify foreground and background features, minimizing the chance of over and under-segmentation. We prove that the negative logarithm of the Dice score lower bounds the feature-distance loss in Lemma 5. This suggests that penalizing feature-distance loss can boost segmentation Dice scores. (See the Appendix for the comprehensive proof)

**Lemma 5.** *Relationship between feature distance loss $\mathcal{L}_{\textit{fd}}$, segmentation Dice score, and constant $k$ for feature vector $F$ derived from image $X$:*

$$-log(Dice \times (k+1)) \leq \mathcal{L}_{\textit{fd}}$$

An increase in the Dice score results in a decrease of the lower bound, which allows for a decrease in $\mathcal{L}_{\textbf{fd}}$. As shown in Figure 1, this relationship justifies the observed correlation between $\mathcal{L}_{\textbf{fd}}$ and the Dice score for all models[1].

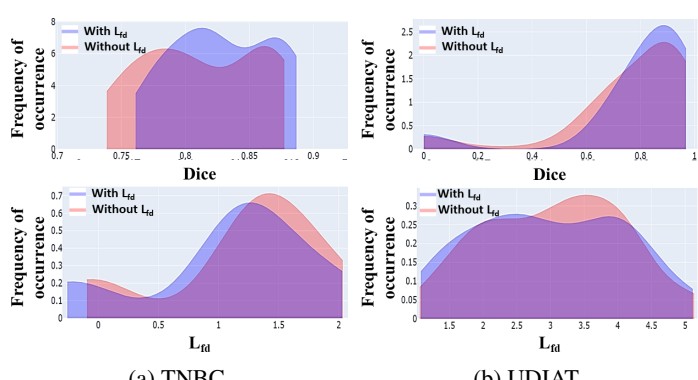

(a) TNBC        (b) UDIAT

Figure 4: Illustration of a right shift and a left shift in the distribution of the test samples with respect to Dice scores and $\mathcal{L}_{\textbf{fd}}$ after the use of $\mathcal{L}_{\textbf{fd}}$ (orange curve).

### 2.3.1 PRACTICAL IMPLEMENTATION OF FEATURE DISTANCE LOSS

**Segmentation Loss $\mathcal{L}_{\textbf{seg}}$.** To penalize spatial prediction, $\mathcal{L}_{\textbf{seg}}$ integrates Dice loss Soomro et al. (2018) and Binary Cross Entropy (BCE) loss Jadon (2020), both essential for image segmentation. These losses evaluate model performance by comparing expected and actual masks. Our technique defines $\mathcal{L}_{\textbf{seg}}$ as a linear combination of Dice and BCE loss, as given in Roy et al. (2024c). For more details, please see the Appendix.
**Layer-wise Feature Distance Loss $\mathcal{L}_{\textbf{fd}}$ and hyper-parameter $\alpha$ regulation.** The U-Net architecture consists of an encoder-decoder structure with skip connections, facilitating the extraction of low-level and high-level features at different spatial resolutions, resulting in multi-scale representations. Implementing a mechanism to penalize feature distance between foreground and background representations at each feature layer is essential for enhancing the model's discriminative power and improving segmentation accuracy. This method promotes the network's ability to learn distinct features at each level, as shown in Fig. 3. A trainable hyper-parameter $\alpha$ is introduced to regulate the importance of each layer in the feature distance loss, with unique $\alpha$ values for each layer. This hyperparameter balances segmentation accuracy $\mathcal{L}_{\textbf{seg}}$ and feature distance loss $\mathcal{L}_{\textbf{fd}}$ at each layer. The experimental section (Section 3.4) will reveal the final $\alpha$ values, indicating each layer's importance in enhancing segmentation scores.
**Warm-Starting $\alpha$.** In the initial model updates, $\alpha$ values are set to zero, optimizing exclusively for $\mathcal{L}_{\textbf{seg}}$

---

[1]Although Lemma 5's bound may not be tight, experiments (Figure 4 and Table 2) show a strict upper-lower bound relationship, indicating that minimizing $\mathcal{L}_{\textbf{fd}}$ directly improves the Dice score.

without factoring in the penalty function $\mathcal{L}_{\mathbf{fd}}$. This method enables $\alpha$ to progressively rise from zero to infinity, consistent with the literature Bertsekas (1997). This approach enables a seamless shift from a constrained to an unconstrained problem, allowing for a thorough exploration of the solution space. Furthermore, starting with a small penalty helps to mitigate potential ill-conditioning associated with large penalties at the outset. We start with $\alpha$ set to zero, permitting the algorithm to iterate multiple times before activating $\alpha$ for training.

## 3 EXPERIMENTS

Section 3.1 outlines the experimental setup, detailing datasets and architectures, and presents a novel dataset for triple-negative breast cancer segmentation. Section 3.2 presents quantitative results, demonstrating improvements in Dice score and IoU due to the inclusion of $\mathcal{L}_{\mathbf{fd}}$. Section 3.3 presents qualitative results comparing generated segmentation masks with ground truth, highlighting enhanced boundary delineation. Section 3.4 presents ablation studies, examining layer-wise performance, the influence of $\mathcal{L}_{\mathbf{fd}}$ on Dice and IoU, and comparisons with state-of-the-art methods.

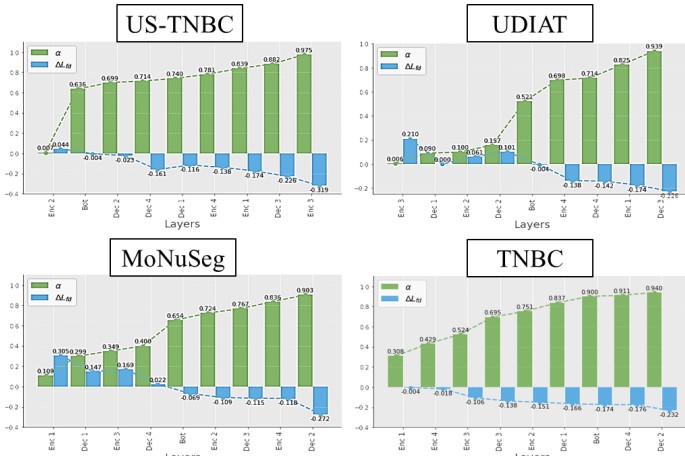

Figure 5: $\alpha$ vs $\mathcal{L}_{\mathbf{fd}}$ for the layers of NucleiSegNet (TNBC and MoNuSeg) and CMUNet (US-TNBC and UDIAT).

### 3.1 SETUP

**Datasets.** We conduct experiments using four datasets. The first is the TNBC dataset (Naylor et al., 2018), which includes histopathology images with high-density glandular tissues and indistinct boundaries, posing challenges for accurate segmentation. Precise TNBC segmentation is vital for detecting and classifying lesions in cancer treatment. The MonuSeg dataset Kumar et al. (2019) consists of Hematoxylin and Eosin-stained histopathology images at 40x magnification. It contains 30 training images with 22,000 annotations and 14 test images with 7,000 annotations, offering a range of tissue types and cell densities for evaluating nuclei segmentation algorithms. We introduce a novel US-TNBC dataset, comprising 15 ultrasound images of TNBC tissues, collected between 2022 and 2023, cropped to a resolution of $721 \times 570$ to retain key anatomical features. Ground truth masks were generated using Fiji, with data anonymized for privacy. The UDIAT dataset Yap et al. (2017) consists of breast ultrasound images, with challenges such as irregular tumor morphology and indistinct boundaries. Additionally, we use an Alzheimer's histopathology dataset for tau protein segmentation Jiménez et al. (2022). The dataset contains two versions, AD $256 \times 256$ (histopathology images with $256 \times 256$ pixels) and AD $128 \times 128$ (histopathology images with $128 \times 128$ pixels). The larger patches ($256 \times 256$ pixels) capture a broader context containing object neighborhood and background pixels, whereas the smaller ($128 \times 128$ pixels) mainly focus on the plaque region. This makes AD $256 \times 256$ more challenging due to more complex background information. The results of AD $128 \times 128$ can be found in the Appendix. Table 1 summarizes these datasets.

**Model Architectures.** Causal mediation and control of $\mathcal{L}_{\mathbf{fd}}$ are independent of neural network architecture. This paper evaluates three prominent UNets and compares the performance of $\mathcal{L}_{\mathbf{fd}}$-penalized models with the latest models. AttentionUNet Jiménez et al. (2022): An enhanced U-Net utilizing gated attention mechanisms that improve segmentation accuracy for small, complex structures by minimizing irrelevant background features. NucleiSegNet Lal et al. (2021) is an architecture developed to address varying nuclei

sizes and overlapping boundaries, employing a robust residual block and attention decoder to enhance object localization and minimize over-segmentation. CMUNet Tang et al. (2023) integrates convolutional layers with a multi-scale attention gate to effectively capture global and local features, addressing the limitations of U-Net in managing the global context. The ConvMixer module integrates features across spatial locations to improve performance.

**Training Details and Evaluation Metrics.**
We employ a 100 epoch training setup for both baselines and $\mathcal{L}_{\mathbf{fd}}$-penalized models. Data augmentation, including flipping and 90° rotations, was utilized for the training set, whereas evaluation occurred on the unaugmented test set (All Samples setup). We used the same augmentation techniques to increase the number of data points for the plots and to select the worst-off and best-off samples. The Adam optimizer was employed with learning rates of 0.0001 for TNBC, MoNuSeg, and UDIAT,

| Dataset | All Samples | Data Type | Worst Off |
|---|---|---|---|
| TNBCNaylor et al. (2018) | 50 | Histopathology | 10 |
| MoNuSegKumar et al. (2019) | 44 | Histopathology | 25 |
| UDIATYap et al. (2017) | 163 | Ultrasound | 35 |
| ADJiménez et al. (2022) | 10k | Histopathology | 500 |
| US-TNBC **(New dataset proposed)** | 15 | Ultrasound | 10 |

Table 1: Summary of datasets. The "All Samples" are the test samples of the dataset while the worst-off samples are the test samples with the lower dice scores.

and 0.001 for AD and US-TNBC. Models are evaluated using Dice Scores and Intersection over Union (IoU) metrics (see Appendix for more details).

## 3.2 QUANTITATIVE RESULTS

| Model | Dataset | $\mathcal{L}_{\mathbf{fd}}$ | Worst Off Samples | | | | Best Off Samples | | | | All Samples | | | |
|---|---|---|---|---|---|---|---|---|---|---|---|---|---|---|
| | | | Dice | $\Delta$ Dice | IoU | $\Delta$ IoU | Dice | $\Delta$ Dice | IoU | $\Delta$ IoU | Dice | $\Delta$ Dice | IoU | $\Delta$ IoU |
| AttnUNet | UDIAT | ✗ | 22.42 | | 29.47 | | 75.86 | | 68.46 | | 67.21 | | 35.61 | |
| | | ✓ | **23.28** | +0.9 | **30.31** | +0.8 | **77.29** | +1.4 | **69.50** | +1.0 | **68.96** | +1.7 | **38.43** | +2.8 |
| | TNBC | ✗ | 77.88 | | 68.64 | | 85.82 | | 74.38 | | 80.61 | | 67.79 | |
| | | ✓ | **77.86** | 0.0 | **68.66** | +0.0 | **86.25** | +0.4 | **77.57** | +3.2 | **81.16** | +0.5 | **69.19** | +1.4 |
| | MoNuSeg | ✗ | 66.03 | | 52.38 | | 82.57 | | 73.48 | | 75.92 | | 61.28 | |
| | | ✓ | **68.61** | +2.5 | **53.06** | +0.7 | **83.62** | +1.0 | **74.50** | +1.0 | **77.97** | +2.0 | **62.87** | +1.6 |
| | AD $256 \times 256$ | ✗ | 56.35 | | 31.92 | | 81.34 | | 70.88 | | 61.14 | | 43.87 | |
| | | ✓ | **57.67** | +1.3 | **33.10** | +1.2 | **85.64** | +4.3 | **72.93** | +2.0 | **64.69** | +3.5 | **46.67** | +2.8 |
| CMUNet | UDIAT | ✗ | 31.56 | | 26.58 | | 90.88 | | 88.25 | | 81.85 | | 69.87 | |
| | | ✓ | **33.19** | +1.6 | **28.17** | +1.6 | **95.32** | +4.4 | **90.01** | +1.8 | **84.22** | +2.4 | **73.02** | +3.1 |
| | US-TNBC | ✗ | 25.08 | | 21.44 | | 86.27 | | 68.09 | | 49.59 | | 34.53 | |
| | | ✓ | **26.94** | +1.9 | **22.35** | +0.9 | **86.04** | -0.2 | **69.35** | +1.3 | **50.22** | +0.6 | **36.52** | +2.0 |
| NuSegNet | TNBC | ✗ | 77.29 | | 68.00 | | 86.49 | | 71.29 | | 81.69 | | 69.22 | |
| | | ✓ | **79.40** | +2.1 | **68.42** | +0.4 | **88.82** | +0.3 | **72.58** | +1.3 | **82.65** | +1.0 | **70.58** | +1.4 |
| | MoNuSeg | ✗ | 63.95 | | 50.05 | | 84.61 | | 70.40 | | 80.95 | | 67.91 | |
| | | ✓ | **64.61** | +0.7 | **52.11** | +2.1 | **84.96** | +0.3 | **71.65** | +1.2 | **81.69** | +0.7 | **68.65** | +0.7 |
| | AD $256 \times 256$ | ✗ | 32.55 | | 23.19 | | 64.75 | | 46.28 | | 51.15 | | 36.17 | |
| | | ✓ | **35.78** | +3.2 | **25.46** | +2.3 | **71.15** | +6.4 | **51.35** | +5.1 | **56.57** | +5.4 | **40.61** | +4.4 |

Table 2: Ablation study on the application of $\mathcal{L}_{\mathbf{fd}}$. The improvement for low dice samples (Worst Off Samples), high dice samples (Best Off Samples), and all test samples (All Samples) can be seen after the application of $\mathcal{L}_{\mathbf{fd}}$. NucleiSegNet Jiménez et al. (2022) is a histopathology segmentation model, so it is not applicable to UDIAT and US-TNBC. Similarly, CMUNet Tang et al. (2023) being an ultrasound segmentation dataset does not apply to training and testing on TNBC. Also, Attention UNet Jiménez et al. (2022) performs poorly (Dice score of 12.96) on the US-TNBC dataset. The changes in Dice and IoU are shown for all three test settings.

The effects of $\mathcal{L}_{\mathbf{fd}}$ are detailed in Table 2, which presents results for all samples, as well as for the Worst-off and Best-off samples based on Dice scores. Table 1 presents the numbers of the best-off and worst-off samples utilized in our experiments. In the case of CMUNet on the US-TNBC dataset, a slight decrease in the Dice score (-0.23) for Best-off samples is offset by improvements in Worst-off samples.

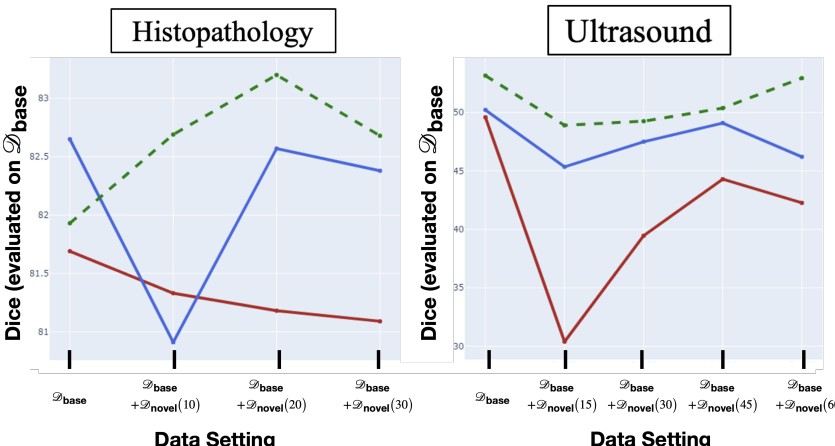

Figure 6: $\alpha$ vs $\mathcal{L}_{\mathbf{fd}}$ for the layers of NucleiSegNet (TNBC and MoNuSeg) and CMUNet (US-TNBC and UDIAT).

On the new US-TNBC dataset, $\mathcal{L}_{\mathbf{fd}}$ results in higher overall Dice scores. The improvements corroborate the theoretical findings in Lemma A.2. (*Takeaway:* Penalizing $\mathcal{L}_{\mathbf{fd}}$ enhances segmentation performance across models and datasets.)

### 3.3 QUALITATIVE RESULTS

Qualitative results for the TNBC, MoNuSeg, AD $256 \times 256$, US-TNBC, and UDIAT datasets are presented in Figures 7 and 8. The red-highlighted areas in the predicted masks without $\mathcal{L}_{\mathbf{fd}}$ indicate segmentation errors, while the green-highlighted regions reflect corrections made by applying $\mathcal{L}_{\mathbf{fd}}$. These experiments illustrate how $\mathcal{L}_{\mathbf{fd}}$ enhances segmentation through boundary refinement and reducing segmentation errors. The resulting masks display sharper, more accurate contours of key structures, preserving fine details and ensuring better anatomical representation. (*Takeaway:* Penalizing for $\mathcal{L}_{\mathbf{fd}}$ results in sharper boundaries, improved detail preservation, and increased consistency in generated segmentation masks.)

### 3.4 ABLATION STUDIES

**Impact of the $\alpha$ Parameter on Feature Distance Loss.** As discussed in Section 2.3.1, $\alpha$ is a trainable parameter that initially starts at zero and regulates the penalty of feature distance loss, $\mathcal{L}_{\mathbf{fd}}$, for each layer of the neural network; the final values of $\alpha$ indicate that the layer with the highest value had the most significant influence on improving the overall dice scores, as shown in Figure 5.

**Comparison with State-of-the-Art Models.** For the TNBC Naylor et al. (2018), UDIAT Yap et al. (2017), and MoNuSeg Kumar et al. (2019) datasets, our method outperforms existing models, achieving Dice score improvements of $+0.96$ (TNBC), $+0.74$ (MoNuSeg), and $+0.75$ (UDIAT) compared to CMUNet Tang et al. (2023) and NucleiSegNet Lal et al. (2021), highlighting the effectiveness of penalizing feature discrepancy in high foreground-background similarity modalities.

**Changes in $\mathcal{L}_{\mathbf{fd}}$ and Dice scores at the sample level.** In Figure 1, a trend between $\mathcal{L}_{\mathbf{fd}}$ and Dice is noted, with some samples exhibiting poor scores in both metrics. Figure 4 presents a frequency plot for $\mathcal{L}_{\mathbf{fd}}$ (orange) and Dice (blue). A shift in $\mathcal{L}_{\mathbf{fd}}$ to lower values and Dice scores to higher values is observed, indicating a significant improvement in Dice scores at the sample level.

| Model | Dice | IoU |
|---|---|---|
| AttnUNet Jiménez et al. (2022) | 80.61 | 67.79 |
| AWGUNet Roy et al. (2024b) | 81.65 | 69.18 |
| GRUNet Roy et al. (2024a) | 80.24 | 66.25 |
| MCFNet Feng et al. (2021) | 73.37 | 57.94 |
| Deep-Fuzz Das et al. (2023) | 77.80 | 64.20 |
| NuSegNet Lal et al. (2021) | 81.69 | 69.22 |
| CellTrp Keaton et al. (2023) | 77.68 | 59.06 |
| ASNet Singha & Bhowmik (2023) | 66.88 | - |
| MMPSO-S Kanadath et al. (2023) | 65.00 | 49.0 |
| Ours | **82.65** | **70.58** |

(a) TNBC Naylor et al. (2018).

| Model | Dice | IoU |
|---|---|---|
| UNet Ronneberger et al. (2015) | 75.00 | 65.00 |
| AttnUNet Jiménez et al. (2022) | 68.96 | 55.00 |
| CMUNet Tang et al. (2023) | 81.85 | 69.87 |
| SCAN Zhang et al. (2020) | 74.00 | 65.00 |
| STAN Shareef et al. (2020) | 78.20 | 69.50 |
| RRC-Net Chen et al. (2023) | 80.40 | 71.81 |
| $EU^2Net$ Roy et al. (2024c) | 83.47 | 72.11 |
| CE-Net Gu et al. (2019) | 72.00 | 61.00 |
| DAUNet Pramanik et al. (2024) | 78.58 | 64.71 |
| Ours | **84.22** | **73.02** |

(b) UDIAT Yap et al. (2017).

| Model | Dice | IoU |
|---|---|---|
| NuSegNet Lal et al. (2021) | 80.95 | 67.91 |
| MedT Valanarasu et al. (2021) | 79.55 | 66.17 |
| HistoSeg Wazir & Fraz (2022) | 75.08 | 71.06 |
| SPPNet Xu et al. (2023) | 79.77 | 66.43 |
| D-Net Islam Sumon et al. (2023) | 73.20 | 58.00 |
| MMPSO-S Kanadath et al. (2023) | 72.00 | 56.00 |
| TSCA-Net Fu et al. (2024) | 80.23 | 67.13 |
| GRUNet Roy et al. (2024a) | 80.35 | 67.21 |
| AWGUNet Roy et al. (2024b) | 79.46 | 66.57 |
| Ours | **81.69** | **68.65** |

(c) MoNuSeg Kumar et al. (2019).

Table 3: Quantitative comparison of segmentation results on different datasets.

# 4 MITIGATING DATASET SHIFTS UNDER ASSUMED EXCHANGEABILITY

Recent studies emphasize the importance of expanding datasets, with particular focus on enlarging medical imaging datasets from multiple sources (Chytas et al., 2024). While initial approaches have leveraged strategies from invariant representation learning to mitigate covariate shifts, current methods are limited, typically addressing only a few covariates at once. The Data Addition Dilemma, introduced by Shen et al. (2024), highlights a critical challenge: in multi-source contexts, increasing the size of training datasets may induce distributional shifts, which paradoxically degrade downstream model performance. Traditional methodologies, based on the assumption of independent and identically distributed (i.i.d.) samples, require adaptation to account for cross-dataset comparisons. In this regard, the introduction of a novel dataset, $\mathcal{D}_{novel}$, alongside a base dataset, $\mathcal{D}_{base}$, poses a significant challenge. Each dataset follows i.i.d. assumptions, but their combination violates this; we address this using exchangeability, a concept extending beyond (i.i.d.) Exchangeability asserts that the joint distribution of a sequence of random variables remains invariant under permutations of indices, a crucial consideration when comparing distinct datasets. By treating $\mathcal{D}_{base}$ and $\mathcal{D}_{novel}$ as part of a sequence of exchangeable random variables, we justify a modified penalty loss function spanning both datasets. The rationale stems from the notion that if the samples from both datasets are indeed exchangeable, then the discrepancy between the foreground feature of a sample from $\mathcal{D}_{base}$ and the background of a sample from $\mathcal{D}_{novel}$ should, in expectation, be comparable to the within-dataset discrepancy observed in the original formulation and vice-versa.

**Definition 6. (Feature Distance Loss under assumed exchangeability):** $F_g(\mathcal{D})/B_g(\mathcal{D})$ *represents foreground/background features from a randomly sampled dataset* $\mathcal{D}$, *which can be either* $\mathcal{D}_{novel}$ *or* $\mathcal{D}_{base}$ *dataset.*

$$\mathcal{L}_{fd}^{exch}(\mathcal{D}_{base} \cup \mathcal{D}_{novel}) = -\log\left(\|F_g(\mathcal{D}_{base}) - B_g(\mathcal{D}_{novel})\|^2 + \|F_g(\mathcal{D}_{novel}) - B_g(\mathcal{D}_{base})\|^2\right) \qquad (2)$$

**Experiments.** We selected TNBC as our base dataset, denoted as $\mathcal{D}_{\textbf{base}}$, using the MoNuSeg dataset as our novel dataset, labeled $\mathcal{D}_{\textbf{novel}}$. We added samples from MoNuSeg sequentially, in batches of 10 images, to $\mathcal{D}_{\textbf{base}}$. All evaluations were performed on $\mathcal{D}_{\textbf{base}}$. Similarly, for the ultrasound datasets, we designated US-TNBC as $\mathcal{D}_{\textbf{base}}$ and UDIAT as $\mathcal{D}_{\textbf{novel}}$, with samples from UDIAT added in batches of 15 images. We compared three methods: a naive method without penalties, a method penalizing for $\mathcal{L}_{\textbf{fd}}$, and a method penalizing for $\mathcal{L}_{\textbf{fd}} + \mathcal{L}_{\textbf{fd}}^{\textbf{exch}}$. Notably, the naive method exhibited a decrease in test set accuracy on $\mathcal{D}_{\textbf{base}}$ as more samples from $\mathcal{D}_{\textbf{novel}}$ were incorporated, consistent with the findings of (Shen et al., 2024). The combination of $\mathcal{L}_{\textbf{fd}} + \mathcal{L}_{\textbf{fd}}^{\textbf{exch}}$ resulted in an overall performance improvement, as illustrated in Figure 6.

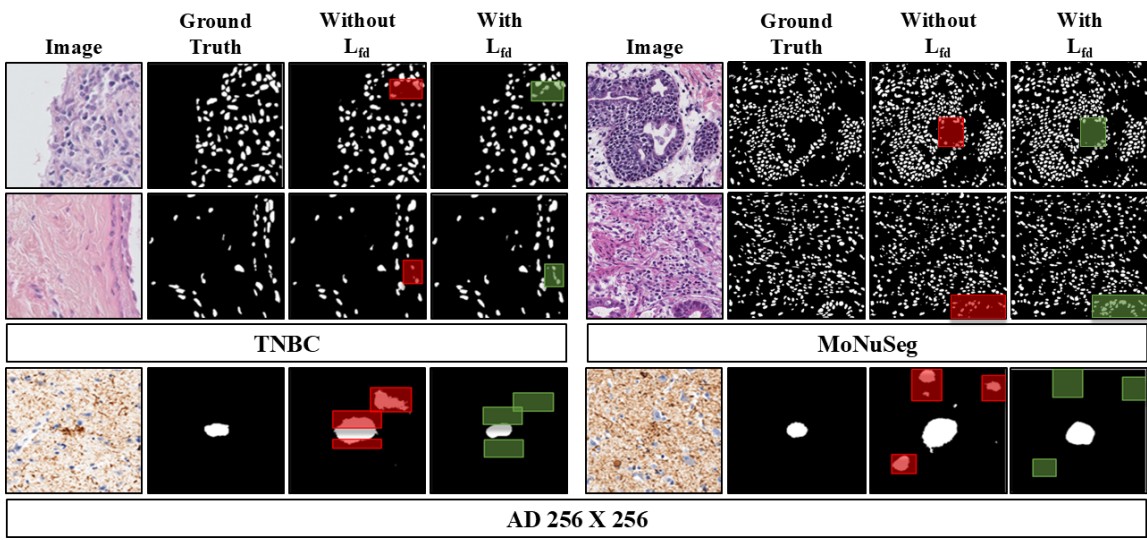

Figure 7: Qualitative analysis of NucleiSegNet for TNBC, MoNuSeg, and $AD256 \times 256$ with $\mathcal{L}_{\mathbf{fd}}$.

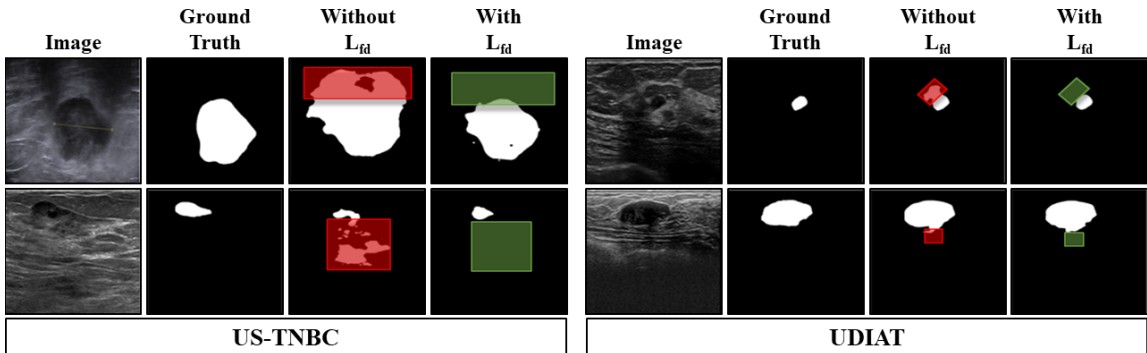

Figure 8: Qualitative analysis of CMUNet for UDIAT and US-TNBC with and without $\mathcal{L}_{\mathbf{fd}}$.

## 5 CONCLUSION

Data scarcity remains a critical challenge in medical imaging deep learning.. Our work addresses this issue by proposing a novel feature discrepancy penalty function that enhances segmentation performance across various modalities. We demonstrate that improved feature separation correlates with higher Dice scores. Through the introduction of a new ultrasound dataset for triple-negative breast cancer, we validate our method across state-of-the-art architectures, achieving competitive results. Our findings also highlight the robustness of our approach against distribution shifts. Future work will explore distribution shift dynamics and the implications of our feature distance penalty on medical image generation tasks.

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

# A  APPENDIX

## A.1  THE SEGMENTATION LOSS

The Dice loss Soomro et al. (2018) and Binary Cross Entropy (BCE) loss Jadon (2020) are crucial for image segmentation tasks, evaluating model performance by comparing predicted and actual masks. The dice loss ($L_{dice}$) and the BCE loss ($L_{bce}$) are defined in Eq. 3 and 4 respectively where $y_{ijk}$ represents the ground truth label for pixel $(i, j, k)$, $\hat{y}_{ijk}$ represents the predicted probability for pixel $(i, j, k)$, $\epsilon$ is a small constant added for numerical stability to avoid division by zero or taking the log of zero, and $N$ is the total number of elements pixels.

$$L_{dice} = 1 - \frac{2 \sum_{i,j,k} y_{ijk} \cdot \hat{y}_{ijk} + \epsilon}{\sum_{i,j,k} y_{ijk} + \sum_{i,j,k} \hat{y}_{ijk} + \epsilon} \tag{3}$$

$$L_{bce} = -\frac{1}{N} \sum_{i,j,k} \Big( y_{ijk} \cdot \log(\hat{y}_{ijk}) \\ + (1 - y_{ijk}) \cdot \log(1 - \hat{y}_{ijk}) + \epsilon \Big) \tag{4}$$

We use a linear combination of $L_{dice}$ and $L_{bce}$ as $L_{seg}$ Roy et al. (2024c). This can be seen in Eq. 5

$$L_{seg} = L_{dice} + L_{bce} \tag{5}$$

## A.2  PROOF

**Lemma 7.** *Relationship between feature distance loss $\mathcal{L}_{fd}$, segmentation Dice score, and constant $k$ for feature vector $F$ derived from image $X$:*

$$-log(Dice \times (k + 1)) \leq \mathcal{L}_{fd}$$

*Proof.* Let $\otimes$ denote element-wise multiplication. From Equation equation 3, we have:

$$\sum_{i,j,k} \tilde{y}_{ijk} = \frac{Dice}{2} \times \frac{\sum_{i,j,k} y_{ijk} + \sum_{i,j,k} \hat{y}_{ijk}}{\sum_{i,j,k} y_{ijk}}$$

Additionally,

$$FD = \frac{\| \sum_k \left( \sum_{i,j} F_{i,j,k} \otimes \tilde{y}_{i,j,k} - \sum_{i,j} F_{i,j,k} \otimes (1 - \tilde{y}_{i,j,k}) \right) \|_2}{\| \sum_{i,j,k} F_{ijk} \|_2}$$

(FD indicates feature distance between the foreground and background features)

We can rewrite $\sum_k \sum_{i,j}$ as $\sum_{i,j,k}$:

$$FD = \frac{\| 2 \sum_{i,j,k} F_{i,j,k} \otimes \tilde{y}_{i,j,k} - \sum_{i,j,k} F_{i,j,k} \|_2}{\| \sum_{i,j,k} F_{ijk} \|_2}$$

Considering the triangle inequality, we get:

$$FD \leq \frac{\|2\sum_{i,j,k} F_{i,j,k} \otimes \tilde{y}_{i,j,k}\|_2}{\|\sum_{i,j,k} F_{ijk}\|_2} + \frac{\|\sum_{i,j,k} F_{i,j,k}\|_2}{\|\sum_{i,j,k} F_{ijk}\|_2}$$

Substituting $\sum_{i,j,k} \tilde{y}_{ijk}$ and rearranging, we get:

$$FD - 1 \leq \frac{\|\sum_{i,j,k} F_{i,j,k} \otimes Dice \times \frac{\sum_{i,j,k} y_{ijk} + \sum_{i,j,k} \hat{y}_{ijk}}{\sum_{i,j,k} y_{ijk}}\|_2}{\|\sum_{i,j,k} F_{ijk}\|_2}$$

Since $\sum_{i,j,k} \tilde{y}_{ijk}$ and $\sum_{i,j,k} y_{ijk}$ are constants during testing, we can consider $\frac{\sum_{i,j,k} \tilde{y}_{ijk}}{\sum_{i,j,k} y_{ijk}}$ as $k'$:

$$FD - 1 \leq \frac{\|\sum_{i,j,k} F_{i,j,k} \otimes Dice \times (1 + k')\|_2}{\|\sum_{i,j,k} F_{ijk}\|_2}$$

$$FD - 1 \leq Dice \times (1 + k')\frac{\|\sum_{i,j,k} F_{i,j,k}\|_2}{\|\sum_{i,j,k} F_{i,j,k}\|_2}$$

Letting $1 - k'$ be a constant $k$, we get:

$$FD \leq Dice \times (k + 1)$$

Taking -log on both sides, we get:

$$-log(FD) \geq -log(Dice \times (k + 1))$$

$$\mathcal{L}_{\mathbf{fd}} \geq -log(Dice \times (k + 1))$$

This completes the proof. $\square$

### A.3   ALGORITHM FOR $\mathcal{L}_{\mathbf{FD}}^{\mathbf{EXCH}}$

---

**Algorithm 1** Loss modification for handing dataset shift in Section 4

---

1: **Input:** Foreground features $F_g$ and background features $B_g$ for each image $i$ in a batch of size $n$
2: **for** each training iteration **do**
3:     **for** $i \leftarrow 1$ to $n$ **do**
4:         $\mathcal{L}_{\mathrm{fd}} = -\log(\|F_{g,i} - B_{g,i}\|_2)$        ▷ penalizing feature distance of foreground and background features for the same image in a batch
5:         $\mathcal{L}_{\mathbf{fd}}^{\mathbf{exch}} = -\log(\|F_{g,i} - B_{g,i+k}\|_2)$ ▷ where $k$ is arbitrary and is introduced after shuffling $F_g$ and $B_g$ of the batch to ensure $F_{gi}$ and $F_{gj}$ are closer to each other y repelling $B_{gj}$
6:         $L_i = \mathcal{L}_{\mathrm{fd}} + \mathcal{L}_{\mathbf{fd}}^{\mathbf{exch}}$
7:     **end for**
8:     loss $\leftarrow \frac{1}{n}\sum_{i=1}^{n} \alpha \times L_i$
9: **end for**
10: **Return:** loss

---

Algorithm 1 outlines the training process for handling dataset shifts from Section 4. To address this challenge, we employ shuffled $\mathcal{L}_{fd}$, which mitigates distribution shifts between the pooled and source datasets by adjusting the feature separation between foreground and background in the shuffled batch. Specifically, the foreground feature of image $i$ ($F_{gi}$) pushes the background feature of image $j$ ($B_{gj}$) for $\mathcal{L}_{fd}^{exch}$, while $F_{gj}$ simultaneously pushes $B_{gj}$ in $\mathcal{L}_{fd}$. This interaction draws $F_{gi}$ and $F_{gj}$ closer, minimizing the distributional shift caused by differences in batch data sources.

## A.4 EXPERIMENTAL SETUP

We developed our segmentation model using Python and implemented it with the TensorFlow and Keras libraries. For data processing, we utilized numpy, OpenCV, and scikit-learn, enabling efficient data handling. We have used the high-performance NVIDIA TESLA P100 GPU to accelerate training and leverage hardware acceleration. The model has been trained for 100 epochs in the initial phase ($\alpha = 0$) and 75 epochs in the second phase with $L_{de}$ ($\alpha \neq 0$). A 5-fold cross-validation was employed for both the baseline and proposed models. A train-test-validation split of 70-20-10% has been applied. Callbacks were used to save the best-performing model during both training phases. To address non-uniform image sizes, all images have been resized to uniform $512 \times 512$ pixels for TNBC Naylor et al. (2018), the newly collected US-TNBC, and $256 \times 256$ for UDIAT Yap et al. (2017) and AD Jiménez et al. (2022) (both $256 \times 256$ and $128 \times 128$). We have applied data augmentation (horizontal and vertical flipping, rotations to the left and right by $90°$) on the training set to train the models and on the test set for increasing the number of data points for the plots. Evaluation of the models has been done on the test set without augmentation.

## A.5 US-TNBC DATASET

The TNBC dataset focuses on Triple-Negative Breast Cancer tissues. The images are typically 721 x 570 pixels in size on average. It consists of 30 images, including 15 ultrasound images and 15 ground truth images. The data collected at baseline includes breast ultrasound images of women aged between 42 and 76 years old. This data was collected between 2022 and 2023, and the images are in PNG format. To make the acquired data useful, some refinement tasks were performed. Firstly, the DICOM images were loaded into a DICOM reader, and the tumor images with-

Table 4: Performance comparison of the SOTA models for $128 \times 128$ and $256 \times 256$ patch images. The best scores are highlighted.

| Model | $128 \times 128$ | | $256 \times 256$ | |
|---|---|---|---|---|
| | Dice | IoU | Dice | IoU |
| UNet Jiménez et al. (2022) | 68.52±0.03 | - | 64.60±0.03 | - |
| AttnUNet Jiménez et al. (2022) | 70.02±0.59 | 56.83±0.99 | 61.14±0.51 | 43.87±0.47 |
| NuSegNet Tomar et al. (2022b) | 72.53±0.41 | 54.47±0.85 | 51.15±0.79 | 36.17±0.17 |
| AttnUNet (Ours) | 71.18±0.65 | **58.11±0.21** | **64.69±0.65** | **46.67±0.21** |
| NuSegNet (Ours) | **74.27±0.45** | 56.51±0.91 | 56.57±0.41 | 40.61±0.41 |

out marking or annotation were selected. Next, the DICOM files were converted into PNG format. The patient information was also eliminated using image cropping software. The images were cropped to retain maximum anatomical information while removing unnecessary boundaries and markers. The ground truth images were generated using Fiji, an open-source image processing program based on ImageJ2. The ground truth masks were produced and then inverted to match the UDIAT dataset mask convention, where the tumor masks are white and the background is black. This dataset is designed to evaluate algorithms for cancer detection, grading, and classification. The steps involved in the collection of the US-TNBC dataset are shown in Fig. 10.

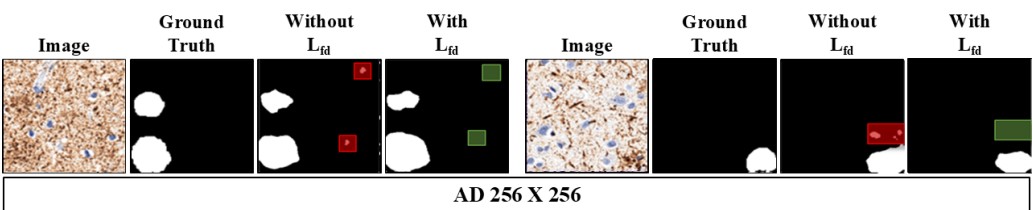

Figure 9: Qualitative analysis of NucleiSegNet for $AD128 \times 128$ with and without $\mathcal{L}_{\mathbf{fd}}$.

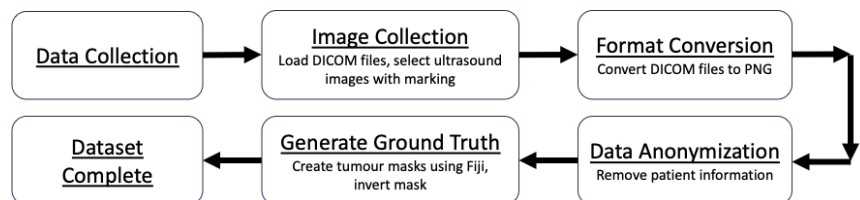

Figure 10: The steps involved in the creation of the US-TNBC dataset.

### A.6 ALZHEIMER'S RESULTS

#### A.6.1 COMPARISON WITH THE STATE OF THE ART

The Alzheimer's dataset by Jimenez et al. Jiménez et al. (2022) consists of fifteen whole slide images containing histological sections from the frontal cortices of patients with AD, provided by the French National Brain Biobank Neuro-CEB. Consent for autopsy and histologic analysis was obtained from the patients or their family members. The AD cases in this cohort exhibit heterogeneity, including variations in tau pathology, staining quality, and tissue preservation. The frontal lobe sections were stained with the AT8 antibody to reveal phosphorylated tau pathology. From the WSIs, at 20x magnification, patches with two levels of context information were generated using an ROI-guided sampling method. Larger patches ($256 \times 256$ pixels) capture a broader context, including the neighborhood and background pixels, whereas smaller patches ($128 \times 128$ pixels) focus mainly on the plaque region without much context information. We keep the experimental setting the same as Jiménez et al. (2022) and evaluate the models on the updated version of the AD dataset. We see an improvement in the performance of the Attention UNet and NucleiSegNet with the use of $\mathcal{L}_{\mathbf{fd}}$ in Table 4.

#### A.6.2 QUALITATIVE ANALYSIS FOR $AD128 \times 128$

Fig. 9 illustrates the improvement in the predicted segmentation mask with the application of $\mathcal{L}_{\mathbf{fd}}$. This showcases the ability of $\mathcal{L}_{\mathbf{fd}}$ to distinguish the highly homogenous distribution of the Alzheimer's histopathology images by penalizing the feature distance between the foreground and background features.

