# OpenReview forum: "Causal Frameworks and Feature Discrepancy Loss: Addressing Data Scarcity and Enhancing Medical Image Segmentation"
_ICLR.cc/2025/Conference — Submitted to ICLR 2025_

### Official Review · Reviewer_dxra · 2024-10-17

**Soundness:** 3
**Presentation:** 4
**Contribution:** 2
**Rating:** 5
**Confidence:** 5

**Summary:**

The paper focuses on addressing data scarcity in medical image segmentation tasks. To me, the contribution of this paper can be viewed as 2 points:
1. A new loss design considered several actual problem settings in medical image segmentation.
2. A new dataset.

**Strengths:**

1. Along with the paper a new ultrasound dataset for triple-negative breast cancer is provided adding value to the paper. While not directly related to the innovation of the method itself, the availability of a new, specialized dataset can be a valuable resource for the research community.
2. The proposed loss considered these:
1>Causal Mediation: The authors introduce the concept of a mediator variable (Z) that captures causally relevant information for segmentation. This mediator is derived from the input image (X) and serves as a proxy for the segmentation target (Y), helping the model learn the true relationship between the image and its segmentation.
2>Feature Distance Calculation: This loss calculates the difference between average foreground and background features in each layer of the network. By minimizing this difference, the model is encouraged to learn more distinct features for the foreground and background, improving segmentation accuracy.
3>Layer-wise Application: This loss is applied to each layer of the U-Net architecture, allowing for fine-grained control over feature separation at different levels of representation. A trainable hyperparameter (α) regulates the importance of loss in each layer, allowing the model to learn the optimal balance between feature separation and overall segmentation accuracy.

**Weaknesses:**

1. I know it is not good to just say lack of novelty, but the segmentation area is developing so fast nowadays. I think the loss design is practical but not eye-attracting at all. The problem setting like data scarcity is something every medical image segmentation paper will mention, and claim their methods will help. The author has made this "data scarcity" in their paper title. I think more experiments showing that this proposed loss is performing well in this "data scarcity" situation should be given at least. Just showing the performance in each dataset is not enough to persuade me. Since the loss can handle data scarcity will contribute to better performance, but better performance does not mean the methods can handle data scarcity. At least more ablation study is needed( ex. Varying Training Data Size, Low-Shot Learning Setup, Comparison with Data Augmentation....)
2. Meta's segment anything is something that changed the game in image segmentation. Why not also give us the score segment anything will perform in this competition.

**Questions:**

I appreciate the practical focus of the paper on addressing data scarcity in medical image segmentation, which is a very relevant challenge. The feature discrepancy loss function seems promising, but I wonder if you could provide more evidence to showcase its effectiveness specifically in data-limited settings, especially given the fast-paced progress in the segmentation field, you need to have something eye-attracting for an ICLR paper.

---

### Official Review · Reviewer_UWQq · 2024-10-18

**Soundness:** 2
**Presentation:** 2
**Contribution:** 2
**Rating:** 3
**Confidence:** 3

**Summary:**

The authors propose a new loss function, termed the “feature distance loss,” to improve semantic segmentation performance. The loss function computes the distance between averaged pixel-level embeddings in the foreground vs. the background at different stages of the neural network. The authors show this loss function improves performance over multiple datasets compared to not including the loss function. Additionally, the authors introduce a new (15 patients) dataset for breast cancer segmentation on ultrasound, which is one of five datasets the authors use to evaluate their methods. Finally, the authors show how their loss function can be extended to improve generalization over datasets collected from different sites.

**Strengths:**

-	The authors show their proposed loss function improves performance over multiple datasets compared to not including the loss function.
-	The loss function is intuitive and can be used with many different architectures, making it easy to use and applicable to many applications.
-	The loss function is extended to improving performance under distribution shifts, which is an interesting extension that shows the flexibility and benefits of the proposed loss.

**Weaknesses:**

-	I find the idea conceptually similar to entropy minimization, which has been a long-standing concept in training deep neural networks for image segmentation [1]. I also find the idea similar to pixel-level contrastive losses for semantic segmentation [2]. Both entropy minimization and pixel-level contrastive losses aim to push apart pixel-level representations of different classes, as do other previous works in semantic segmentation (e.g., [3]). The proposed work takes an alternative angle at this idea—by encouraging foreground/background classes to be far apart at many layers of the network—but a discussion of how the proposed idea relates to similar historical ideas in semantic segmentation would be appropriate.
-	Evaluation focuses primarily on comparing networks trained with the loss function vs. without the loss function. While this is convincing for showing that the proposed loss function improves performance, it does not give a good understanding of how the proposed loss function compares to other loss functions or approaches developed to deal with limited labeled data. As a result, I do not have a sense for how impactful the proposed loss function is.
-	Minor formatting note: the authors appear to use \citet instead of \citep, resulting in in-text citations where they are meant to be in parentheses.
-	Minor note on framing: the Introduction focuses on breast cancer, however the paper and contributions are not particularly focused on breast cancer. You evaluate the loss function over multiple datasets (breast cancer and otherwise), and the primary contribution of the paper is a loss function that can be applied to any semantic segmentation network. Further, you show that the loss function can be extended to solve other problems in semantic segmentation, such as handling distribution shifts. Perhaps an alternative framing that focuses on the loss function and its applications would be more appropriate.
-	Minor note: In the title you refer to your loss as “feature discrepancy loss,” while in the main text you refer to your loss as “feature distance loss.” Staying consistent would help the reader.
-	Minor note: Figure 6 seems to be missing a legend.

In summary, I find the loss function useful and believe it improves performance. However, it is conceptually similar to past loss functions that have been developed and it is not evaluated well enough for me to be convinced of its impact. Given the incremental nature of the contribution and the minimal evaluation, I recommend to reject.

[1] Grandvalet, Yves, and Yoshua Bengio. "Semi-supervised learning by entropy minimization." Advances in neural information processing systems 17 (2004).

[2] Chaitanya, Krishna, et al. "Local contrastive loss with pseudo-label based self-training for semi-supervised medical image segmentation." Medical image analysis 87 (2023): 102792.

[3] He, Xingjian, et al. "Consistent-separable feature representation for semantic segmentation." Proceedings of the AAAI Conference on Artificial Intelligence. Vol. 35. No. 2. 2021.

**Questions:**

-	Can the loss function be extended to multi-class settings?

---

### Official Review · Reviewer_QXYK · 2024-10-28

**Soundness:** 2
**Presentation:** 3
**Contribution:** 1
**Rating:** 3
**Confidence:** 4

**Summary:**

This paper presents a causal-inspired foreground-background feature discrepancy penalty function aimed at enhancing medical image segmentation tasks. The method, grounded in a causal framework, demonstrates strong performance across multiple medical image segmentation challenges.

**Strengths:**

1.	The attempt to improve the medical image segmentation from a causality’s perspective is interesting.
2.	The method achieve sota performance on multiple datasets with different modalities.
3.	The paper is well-written and easy to follow.
4.	The paper introduce a new ultra-sound dataset for triple-negative breast cancer.

**Weaknesses:**

1.	The theoretical basis on causal inference appears disconnected from the method itself. The only connection is the acknowledgment that segmentation results rely on latent representations, highlighting the need for their improvement—an idea that is widely recognized. Therefore, reiterating this from a causal perspective seems unnecessary. Additionally, the propositions mentioned in the paper, such as functional relationships and conditional independence, do not seem relevant to the method.
2.	The design of the method appears trivial, as it merely adds a loss term at intermediate levels to increase the distance between foreground and background representations. Many previous works have explored similar concepts, albeit in different forms, including deep supervision [1], local contrastive learning [2], and learning-to-rank [3].
3.	There is some ambiguity in the methodology (see questions).

[1] Dou, Qi, et al. "3D deeply supervised network for automatic liver segmentation from CT volumes." Medical Image Computing and Computer-Assisted Intervention–MICCAI 2016: 19th International Conference, Athens, Greece, October 17-21, 2016, Proceedings, Part II 19. Springer International Publishing, 2016.

[2] Chaitanya, Krishna, et al. "Contrastive learning of global and local features for medical image segmentation with limited annotations." Advances in neural information processing systems 33 (2020): 12546-12558.

[3] Gong, Shizhan, et al. "Segmentation of Tiny Intracranial Hemorrhage Via Learning-to-Rank Local Feature Enhancement." 2024 IEEE International Symposium on Biomedical Imaging (ISBI). IEEE, 2024.

**Questions:**

1.	The $\tilde{y}$ in definition 4 seems to be predicted masks. Why not use the ground truth mask but instead use a predicted mask with error for supervision? As the size of the feature map is smaller than the final mask, how do you define y for feature patches containing both foreground and background pixels?
2.	What is k in definition 4? Is it channel or the depth dimension? It seems you are using 2D images for illustration but in Appendix you mention pixel (i,j,k), this is confusing.
3.	The formula for $B_g$ seems to be wrong. Should it sum over k?
4.	It is better to replace k with another symbol in Lemma 5. Will cause confusion with the k in definition 4.
5.	The proof of Lemma 5 does not consider the difference of sizes between feature maps and images.
6.	Will the TNBC data be released?

---

### Official Review · Reviewer_9hEf · 2024-11-01

**Soundness:** 1
**Presentation:** 2
**Contribution:** 2
**Rating:** 3
**Confidence:** 4

**Summary:**

This paper presents a new approach to address data scarcity in medical image segmentation using a causal-inspired feature discrepancy (FD) loss function. Theoretical analysis has been provided to demonstrate that the FD loss could improve the segmentation performance in Dice score by enhancing the separation of foreground and background feature. The paper introduces a new ultrasound dataset for triple-negative breast cancer and demonstrates the effectiveness of the approach across various state-of-the-art architectures. Additionally, it shows the robustness of the method against distribution shifts when integrating datasets from multiple sources. The key contributions include a mathematically grounded loss function, a new dataset for TNBC, and strategies to mitigate dataset distribution shifts.

**Strengths:**

The paper introduces a novel causal perspective of image segmentation with scarce labeled data and points out why current semi-supervised and augmentation-based methods are inadequate for resolving the data scarcity issue.
The paper has mathematically proved that the proposed feature discrepancy loss could improve the upper bound of the Dice score.

**Weaknesses:**

The central claims of the paper are adequately supported with evidence. It is said In Sec. 2.2 that the paper addresses the issue through causal mediation. However, the connection between the proposed feature discrepancy loss and causal medication is not explained clearly. The empirical study and theoretical analysis of the feature discrepancy loss only lead to the conclusion that imposing label supervision on intermediate features could help improve the Dice score, which has already been validated in the previous work (Isensee et al. Nature Methods, 2021)

**Questions:**

Suggestion:
To support the claim that the feature discrepancy loss improves the estimation of the mediate variable Z, it is necessary to include visual or quantitative analysis of the feature since it is assumed that "mediator variable Z must capture causally relevant information for segmentation while discarding irrelevant or spurious correlations". It would be better if theoretical analysis could be provided to demonstrate how the proposed loss leads to causal relevance or irrelevance.

---

### Meta-Review · Area_Chair_gan8 · 2024-12-16

**Metareview:**

This work aims to address data scarcity and enhance medical image segmentation via a causal perspective with limited labeled data. The proposed method achieves SOTA performance on several segmentation datasets. Moreover, the authors proved their statement via mathematical proof.

This paper received 3x reject and 1x marginally below the acceptance threshold ratings from reviewers. The major concerns raised by reviewers centered around the limited clarity of methodology and the ambiguous connection between proof and the method itself. However, the authors did not participate in the discussion phase or improve their manuscript.

Given the consensus of reviewers, rejection is recommended.

**Additional Comments On Reviewer Discussion:**

Reviewers raised several valuable comments for improving the paper, including clarifying the connection between mathematical proof and the method itself, explaining more details of loss design and introducing its novelty, as well as incorporating more baselines such as Segment Anything, etc. However, the authors were not included in the discussion and no response has been made.

---

### Decision · Program_Chairs · 2025-01-22

Reject